# Phase-Optimized Peristaltic Pumping by Integrated Microfluidic Logic

**DOI:** 10.3390/mi13101784

**Published:** 2022-10-20

**Authors:** Erik M. Werner, Benjamin X. Lam, Elliot E. Hui

**Affiliations:** Department of Biomedical Engineering, University of California, Irvine, CA 92697, USA

**Keywords:** pneumatic logic, droplets, microfluidics, micropump, point-of-care

## Abstract

Microfluidic droplet generation typically entails an initial stabilization period on the order of minutes, exhibiting higher variation in droplet volume until the system reaches monodisperse production. The material lost during this period can be problematic when preparing droplets from limited samples such as patient biopsies. Active droplet generation strategies such as antiphase peristaltic pumping effectively reduce stabilization time but have required off-chip control hardware that reduces system accessibility. We present a fully integrated device that employs on-chip pneumatic logic to control phase-optimized peristaltic pumping. Droplet generation stabilizes in about a second, with only one or two non-uniform droplets produced initially.

## 1. Introduction

Over the past decade, droplet microfluidic technologies have enabled many new research and commercial applications [1,2,3,4]. They offer low reagent consumption, fast mixing times, and can be scaled for high throughput applications [5]. Precise control of droplet volume and composition are key to the power of droplet microfluidic systems [6]. Predominantly, droplets are generated passively using the physical design of the microfluidic device and fluid flow supplied by a syringe pump or pressure controller. Such systems exhibit a stabilization period upon flow initiation, lasting 1 to 3 min, characterized by higher variation of droplet volume [7,8]. In many applications, the initial droplets can simply be discarded, but loss of material may not be acceptable for precious samples. One example is the use of droplet microfluidics to create patient-derived organoids for personalized cancer therapy [9]. Here, the initial biopsy must be expanded in organoid culture to produce adequate cell numbers to support a set of patient-specific drug sensitivity assays, and the loss of biopsy material can delay drug screening beyond the window for making clinical treatment decisions.

Droplet stabilization time can be sharply reduced or eliminated by employing active generation strategies such as microvalves [10], acoustics [11], laser cavitation [12], or automated aspiration [13]. One particularly robust strategy is to employ antiphase peristaltic pumping in which the dispersed and continuous phases are alternately driven by phase-shifted gears rolling along flexible tubing [7]. These approaches are effective but require significant off-chip machinery, increasing cost, complexity, and size, and making broad dissemination more challenging.

In this work, we implemented phase-optimized peristaltic pumping by using an integrated control system on a single monolithic chip. Microfluidic digital logic, created with a network of pneumatic valves and channels, is used to coordinate vacuum pressure signals that drive a set of peristaltic pumps built into the device and control all fluid handling. The device includes four phase-synchronized peristaltic pumps, a serpentine mixer, a T-junction droplet generator, and an area for droplet observation (Figure 1). It is made from inexpensive readily available materials and a rapid prototyping method using a desktop CO_2_ laser allowed us to quickly optimize the device design. The integrated peristaltic pump draws liquid from external reservoirs, and the logic circuitry is powered by a vacuum pressure source which can be manually generated. Electricity and additional control equipment are not required, making the system suitable for usage in a wide variety of settings.

## 2. Materials and Methods

### 2.1. Laser Calibration

Prior to device fabrication, a calibration procedure was completed using the laser to enable precise control of microfluidic channel depth. 1-mm square calibration patterns were cut using raster mode at 1000 PPI with a CO_2_ laser (VLS 4.6, Universal Laser Systems, Scottsdale, AZ, USA) into a sheet of 1/16” mm PMMA (1227T769, McMaster-Carr, Elmhurst, IL, USA). Patterned surfaces were profiled using a 3D laser scanning microscope (VKX-110, Keyence, Itasca, IL, USA), and a linear regression on the average depth vs laser power was used to determine optimal laser power and speed settings for device fabrication (Appendix A).

### 2.2. Chip Fabrication

Microfluidic logic circuits were fabricated by aligning and stacking layers of patterned PMMA and silicone film (HT-6240, Rogers Corp, Chandler, AZ, USA) similar to previous work [14]. Device features were designed using commercial CAD software (AutoCAD 2020 for Mac, Autodesk, San Rafael, CA, USA, accessed on 21 May 2020) and transferred to the CO_2_ laser using its printer driver. All fluidic channels were machined into PMMA substrates by using raster mode with controlled widths and depths ranging from 200 µm to 1000 µm into PMMA substrates. A vector mode was used to cut completely through either the PMMA or silicone film. All features were processed at 1000 PPI using laser power and speed settings determined by the calibration procedure. Following patterning, parts were cleaned using deionized water and dried with compressed air. Clean and dry parts were assembled by stacking each layer over stainless steel tapered dowel pins (90681A002, McMaster-Carr), fitting alignment holes included at each corner of the device design over the small end of each pin. Following assembly, devices were compressed and baked to seal all layers together. Additional pieces of silicone film were placed on either side of the device to function as padding (Appendix A). The assembly was clamped between glass microscope slides using office binder clips and placed sideways into an oven (DX300, Yamato Scientific, Santa Clara, CA, USA) for two hours at 110 °C to produce a bubble-free, air-tight, reversible seal between the layers. Devices were allowed to cool completely before unclamping and use. Complete protocols for chip design, fabrication, and assembly can be accessed online at protocols.io [15].

### 2.3. Valve Volume Measurement

The movement of an air–liquid interface in a channel downstream of a valve was acquired using a USB microscope (Opti-Tekscope, Chandler, AZ, USA), while the actuation pressure applied to the valve was measured using a pressure sensor (Honeywell TruStability HSC, Charlotte, NC, USA). A computer vision program was written using MATLAB (MATLAB r2021a, The MathWorks Inc, Natick, MA, USA, accessed 17 May 2021) and used to process each video frame and extract the area of the channel containing liquid. Pixel area was converted to volume using the channel cross section and a ruler that was laser-cut into the device as a reference length.

### 2.4. Oscillator Pump Operation

Oscillator pumps consisted of three or five inverter gates connected in a ring to form a ring oscillator. Outputs from three or four of the inverter stages were each further connected to liquid handling valves arranged in a series so that sequential actuation of each valve resulted in peristaltic pumping (Appendix A). Water-in-oil droplets were produced using deionized water and food dye (McCormick, Hunt Vally, MD, USA) as the aqueous phase reagents, and pure corn oil (Mazola, Oakbrook Terrace, IL, USA) as the continuous phase.

Vacuum power for oscillator pumps was supplied either from an electric vacuum pump (4176K11, McMaster-Carr, Elmhurst, IL, USA) via a miniature vacuum pressure regulator (V-800-30-W/K, Coast Pneumatics, Anaheim, CA, USA) or directly from a disposable 60 mL syringe (VacLoc, Merit Medical, South Jordan, UT, USA). For response time measurements, input vacuum was switched between 70 kPa and atmospheric pressure using a miniature 3-way solenoid valve (S10, Pneumadyne, Plymouth, MN, USA) connected to a microcontroller (ATMega 328p, Microchip, Chandler, AZ, USA). Pressure measurements were acquired at 50 Hz using a pressure sensor (PX139, Omega Engineering, Norwalk, CT, USA) connected to the microcontroller. Reagents were supplied from Falcon tubes or a 96-well plate through short lengths of tubing (Tygon E-3603, Cole Parmer, Vernon Hills, IL, USA) attached to a laser cut PMMA manifold and held in place with a PDMS gasket and office binder clips.

### 2.5. Oscillator Pump Waveform Analysis

A video of the incident light reflected from pump valves was imaged at 1080p and 240 FPS (iPhone XR, Apple Computer, Los Altos, CA, USA) and processed with a custom program written using OpenCV [16]. For each pump valve, a region of interest (ROI) was defined around the area of reflected light. The average pixel intensity for each ROI was extracted for each frame of video. Data were analyzed using the normalized average pixel intensity inside each ROI.

### 2.6. Droplet Analysis

Videos of droplet generation were acquired at 1080p and 20 FPS using a DSLR camera and macro lens (EOS Rebel T1i, Cannon, Tokyo, Japan). Custom software written using OpenCV [16] was used to process each video frame and extract the contour of each droplet. Droplet volumes were calculated by measuring the area of each contour, fitting a capsule to each contour, and converting voxels to liquid volume using a ruler laser-cut into the device as a reference length. Flow rate measurements were calculated using the displacement of the centroid of each contour in each frame and the cross-sectional area of the channel.

## 3. Results

### 3.1. Integrated Controller Design

We implemented our microfluidic digital logic controller using normally closed elastomeric membrane valves similar to those first described by Grover and Mathies [17]. Valves were fabricated by sandwiching an elastomeric membrane between two layers of complementary microfluidic channels. The application of vacuum pressure to one side of the membrane opens the valve, connecting the channels on the opposite side (Figure 1a). Vacuum-powered digital logic gates were created by connecting valves though channel networks of varying fluidic resistances. Our laser cut inverter gates exhibited a sharp non-linear transfer function (Figure 1b), indicating they are well-suited for producing robust logic circuits [18]. Compared to other microfluidic logic technologies [19], the three-layer composition of normally closed logic gates makes them relatively simple to fabricate. Notably, they can also connect directly to liquid handling valves to control fluid flow without requiring any additional components or fabrication steps, making them appropriate for integrated microfluidic control.

For our integrated controller, a ring oscillator was constructed by connecting an odd number of inverter gates in a closed loop [18]. Pumps were created by attaching liquid handling valves to the pressure outputs of the ring oscillator in order to generate a peristaltic pumping pattern (Figure 1c). A ring oscillator with N inverters will generate a cyclic sequence of 2×N unique states, transitioning from each state to the next after a period of time that can be adjusted by tuning the source vacuum pressure, channel geometry, and valve switching speed [18]. Connecting at least three of these outputs to liquid handling valves generated the sequential squeezing pattern required for peristaltic fluid pumping. While this pump controller design is straightforward in theory, in practice it was difficult to synchronize multiple pumps. We experimented with many design variations before our device produced monodisperse droplets. Our final optimized design (version 32) incorporated four oscillator-driven peristaltic pumps operating in parallel, two for aqueous reagents, and two for disperse phase reagents, to generate water-in-oil droplets (Figure 1d).

### 3.2. Rapid Prototyping

Since predicting and modeling the behavior of compressible fluid flow in complex networks is difficult, a desktop CO_2_ laser fabrication process was developed to rapidly and inexpensively optimize our integrated microfluidic digital logic control circuits. Laser engraving of plastics has previously been shown to be an effective method for rapid prototyping of microfluidic channels [20,21,22]. Laser engraving has also been demonstrated for making microfluidic logic circuits, but required a chemical bonding step that our process does not require [23]. Compared to the previous methods of photolithography or wet etching, our process does not require any hazardous chemicals, specialty materials, or a clean room (Appendix A). As a result, circuits could be prototyped quickly and easily. A complete design-build-test-learn cycle could be finished in as little as one day, a rate that was sustainable for several days at a time. The rapid turnaround time allowed us to explore many different design ideas and quickly produce an optimized device (Appendix A). In total, over 30 design iterations were fabricated and tested during the prototyping process.

Further, we leverage the ability of the laser to easily and independently control channel width and depth (Appendix A) to produce channels of a wide range of hydraulic resistances. In this work, inverter gate pull-up resistor channels were designed using 200 µm × 100 µm (W × H) channels to provide high resistance, while channels connecting the inverters were enlarged to 400 µm × 400 µm and channels distributing power were further enlarged to 1000 µm × 1000 µm. By using multiple depths, high resistance channels provided over 3000 times as much resistance per millimeter (Appendix A). In the context of microfluidic logic circuit design, this allowed us to reduce the surface area required for resistors and provided the flexibility to place gates anywhere on the device with minimal pressure drop from resistive losses in power and interconnect channels. A similar approach using multiple channel depths to vary the resistance of pneumatic circuits has been previously employed in photolithographic and micro milling processes, but required challenging and time consuming alignment of photomasks [24] or delicate tools [14]. In comparison, a desktop CO_2_ laser is easy to align and can produce circuits that approach the density of higher resolution methods (Appendix A).

### 3.3. Droplet Generator Optimization

Typically, peristaltic pumps are not well suited for droplet applications as they produce pulsatile flow, and most droplet generators are sensitive to changes in flow rate. In recent work, it was demonstrated that synchronized peristaltic pumping can generate well-controlled droplets that are robust to changes in instantaneous flow rate [7]. Our device uses a similar approach but employs an integrated controller to synchronize multiple embedded peristaltic pumps.

While an oscillator circuit has previously been used to control one peristatic pump [18], we observed that controlling multiple pumps from the same oscillator resulted in nonuniform pump performance. When two or more pumps were connected to an oscillator circuit in parallel, the pumps closer to the oscillator produced higher flow rates. This can be explained by the compressible flow governing pneumatic circuits [18]. The pump valves farthest from the vacuum sources are expected to actuate the slowest due to the additional capacitance and resistance of the channels that connect them. Several design modifications were tested to improve pump strength and uniformity to ensure consistent mixing and droplet generation. Initially, pump designs were changed to vary the flow rates of the aqueous and disperse phases and evaluated qualitatively based on the ability of the pumps to self-prime, evenly mix the two aqueous reagents, and produce droplets. In later revisions, the uniformity of the pumps was optimized to generate monodisperse droplets. Droplet monodispersity, mixing (Appendix A) and valve sequencing were quantified from video recordings. A selection of significant pump design revisions and their changes are presented in the supplemental materials (Appendix A). During the prototyping process, four key changes were identified that yielded improved droplet uniformity.

First, pump valves were placed in series (Appendix A) with the inverters of the oscillator (versions 18 and later). Routing the oscillator signal through the pump valves ensured that the signal arrived to each pump valve before reaching the next inverter in the oscillator and produced more consistent pumping across multiple pumps. This arrangement allowed a single oscillator circuit to control and synchronize four pumps operating in parallel, reducing the surface area required for control circuitry.

Second, the pump was changed from a “double chamber” configuration [25], where each pump cycle was driven by two valves closing, to a “single chamber” pumping configuration, where each pump cycle was driven by a single valve closing (versions 27 and later). While in some circumstances the double chamber pumping pattern can produce more efficient pumping [25], it generates two forward flow pulses per pump cycle, resulting in different sized droplets from each pulse (Figure 2a,b). We found that configuring the oscillator pump to produce one droplet per pump cycle resulted in the most uniform droplet production (Figure 2c,d).

Third, fluidic capacitors were added before and after each stage of pump valves (version 32). Fluidic capacitors were created by adding extra volume to the channels connecting the inverter outputs to each stage of pump valves. The added volumes introduced additional delays between inverter valve and pump valve actuation and prevented pump valves from opening before the previous pump stage had closed (Figure 3). This made the pumps stronger and more robust by eliminating the possibility that all pump valves could be open simultaneously.

Finally, the depth of the channels directly underneath the valve seats was reduced to 150 μm (versions 30 and later). This limited the maximum deflection of the membrane and minimized changes in the stroke volume of the pump when higher levels of vacuum were applied (Appendix A). Previously, channel depth was limited to 50 μm using a wet etching fabrication process. While channels produced with the laser fabrication process are generally larger, the ability to control the depth allowed us the flexibility to use many channel depths without complicating the fabrication process. In future designs, using deeper channels may also be useful for increasing the maximum flow rate of oscillator pumps.

Throughout the prototyping process, several additional strategies for improving pump strength and consistency were tested but ultimately not selected in the optimized design. Additional oscillator circuits with pumps were added but required synchronization circuitry to produce monodisperse droplets. Check valves [26] were added to prevent backwards flow but required at least two pumps in parallel to produce enough pressure to open the valves and did not improve droplet monodispersity as the pumps were still not synchronized. Finally, a fourth row of pump valves was added to a five-inverter oscillator design to prevent backwards flow when all pump valves were open simultaneously. While this improved pump strength similar to the addition of capacitors, a “single chamber” pump produced droplets of more consistent size with a simpler controller and this design was not necessary.

### 3.4. Droplet Generator Analysis

The optimized oscillator pump controller (version 32) produced droplets with excellent mono-dispersity (CV < 2%) when operated at a constant vacuum pressure (Appendix A) and functioned across a range of operating pressures from 25 kPa and greater (Figure 2c). The pumps were self-priming, and the device could produce droplets at rates between 1 Hz and 3 Hz continuously for over 24 h. For all devices, we observed that droplet volume decreased as input vacuum pressure increased, although at higher vacuum levels (>50 kPa) the change in volume was minimal. This can be attributed to the increase in peak flow rate produced by the pump system when supplied with increased vacuum pressure (Figure 2d). At higher vacuum levels the pump valve membrane is expected to deflect more, displacing a greater volume of fluid per stroke, and producing a higher peak flow rate. In our system we observed peak flow rates ranging from 38.9 μL/min at 25 kPa operating vacuum to 60.33 μL/min at 80 kPa operating vacuum, corresponding to peak velocities from 3.60 mm/s to 5.59 mm/s. With approximately equal flow rates of the disperse and continuous phases, we expect a maximum capillary number from 1.60 × 10^−3^ to 2.48 × 10^−3^ (Appendix A) and droplet breakoff to occur in the squeezing regime [27]. This observation is consistent with previous studies that have demonstrated smaller droplets are produced with increasing capillary numbers [28]. While our optimization efforts were able to improve droplet mono-dispersity at all operating pressures, the deflection of the valve membrane and thus pump stroke volume still ultimately depended on the operating pressure. This prevented the system from producing droplets of uniform size across its entire range of operating pressures and oscillator frequencies.

In contrast to electronic control apparatus, the integrated controller relies on vacuum pressure to drive the flow of air through its circuits. Due to the small channel dimensions, the flow rate of air through the control circuitry is also small and permits operation from a small portable vacuum reservoir such as a handheld disposable syringe [18], analogous to battery-operated low-power electronics. Pulling and locking the plunger of a 60 mL syringe, the system yielded 465 droplets with a cumulative CV < 5% over about 3 min of operation before pressure loss eventually caused the droplet volume to drift. (Figure 4a, Appendix A). In the first 200.6 s of operation, droplet volumes ranged from 16.8 nL at 90 kPa to 18.0 nL at 62 kPa. Connecting the syringe via a vacuum pressure regulator set to 50 kPa helped to stabilize the slow decay of the vacuum supply, but vacuum loss through the vent of the regulator ultimately reduced the total time the system could maintain pressure to about 1 min and the system only produced 158 droplets before the CV exceeded 5% (Figure 4b). Electricity free operation of microfluidic oscillator circuits for longer periods of time has been previously demonstrated using a manual vacuum pump with an integrated reservoir [18] or a microfluidic venturi tube [29].

Due to the tight integration of the embedded controller, the system also responds quickly to changes in applied vacuum pressure. Upon application of a 70 kPa vacuum pressure, the device transitioned from resting to producing monodisperse droplets in approximately 1 s (Appendix A). During this transition period, one or two non-uniform droplets were produced as the system expelled disperse phase remaining in the T-junction from the previous operation cycle, producing about 100 nL of waste. The fast response time and short stabilization time of this system compares favorably with syringe pump droplet generators, which can take several minutes to stabilize [8]. Rapid stabilization reduces waste when encapsulating precious samples into droplets [9]. Similar to previously reported peristaltic droplet systems [7], the fast response time of this device may be useful for intermittent measurement or long term monitoring applications.

## 4. Discussion

We have demonstrated a novel microfluidic droplet generator system consisting of a series of peristaltic pumps controlled by an integrated pneumatic logic circuit. The system produces monodisperse droplets with volumes ranging from 36.8 nL to 16.8 nL upon application of a constant vacuum pressure between 25 kPa and 80 kPa. The droplet volume changes minimally as long as supply pressure is maintained above 50 kPa. This allows generation of monodisperse droplets for a short time using a portable electricity-free vacuum source such as a disposable syringe. Minimal reagent waste is produced due to a short stabilization time of about 1 s. The integrated controller allows reagents to be consumed directly from a well plate (Appendix A), eliminating the swept volume introduced by syringes and tubing. The design and optimization of the integrated controller was enabled by a rapid prototyping method developed using a desktop laser cutter and low cost materials, enabling the use of this powerful control technology in a greater number of lab on a chip applications.

In future work, integrated digital microfluidic controllers such as the one used in this droplet system offer the possibility to automate sequences of fluid handling operations. While our device generated droplets using pumps synchronized to a single oscillator circuit, alternate pump patterns could be achieved by reorganizing the oscillator configuration or incorporating additional circuitry such as memory, counters, and shift registers [30] to automatically generate repeating patterns of droplets such as dilution series, applicable in a diverse array of bioanalytical applications [31,32,33,34]. The integrated and portable nature of the system make it well suited for applications such as environmental monitoring, wearable heath monitoring, and point-of-care diagnostics. The low reagent consumption may be useful in situations when reagent material is limited, such as emulsifying patient-derived samples. As the system is constructed of entirely PMMA and silicone, it can be made disposable or recyclable and is inexpensive to produce. Finally, it may be used in environments where ionizing radiation, corrosive gases, or strong magnetic fields may hinder the use of electronics, such as in outer space, or inside an MRI scanner [35,36]. Integrated controllers may support new applications for autonomous, standalone droplet microfluidic systems that can be broadly deployed at low cost.

## Figures and Tables

**Figure 1 micromachines-13-01784-f001:**
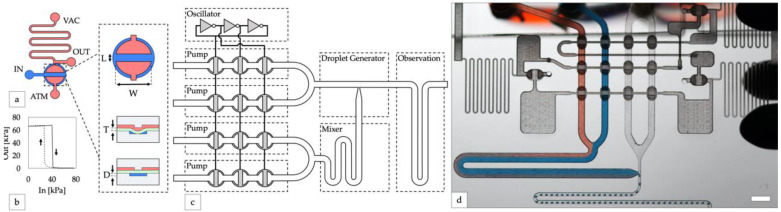
Design of integrated controller for droplet generator. (**a**) An inverter gate is created by attaching high and low resistance channels to opposite sides of a valve. Channels on the top and bottom layers are shown in red and blue, respectively. A top-down valve detail shows dimensions of the valves used in this study (length L = 0.5 mm, width W = 1 mm). Cross-section valve detail shows an open valve (upper) and closed valve (lower) with elastomeric membrane in green (thickness T = 250 µm) and depth of the displacement chamber (depth D = 150 µm). (**b**) The transfer function of a laser-cut inverter gate produces a sharp non-linear switch in output vacuum as input vacuum is increased (solid line, valve opening) and decreased (dashed line, valve closing). (**c**) Block diagram of the droplet system. A ring oscillator created from inverter gates synchronizes four peristaltic pumps. (**d**) Photo of the device generating water-in-oil droplets containing blue and red dye (scale bar = 3 mm).

**Figure 2 micromachines-13-01784-f002:**
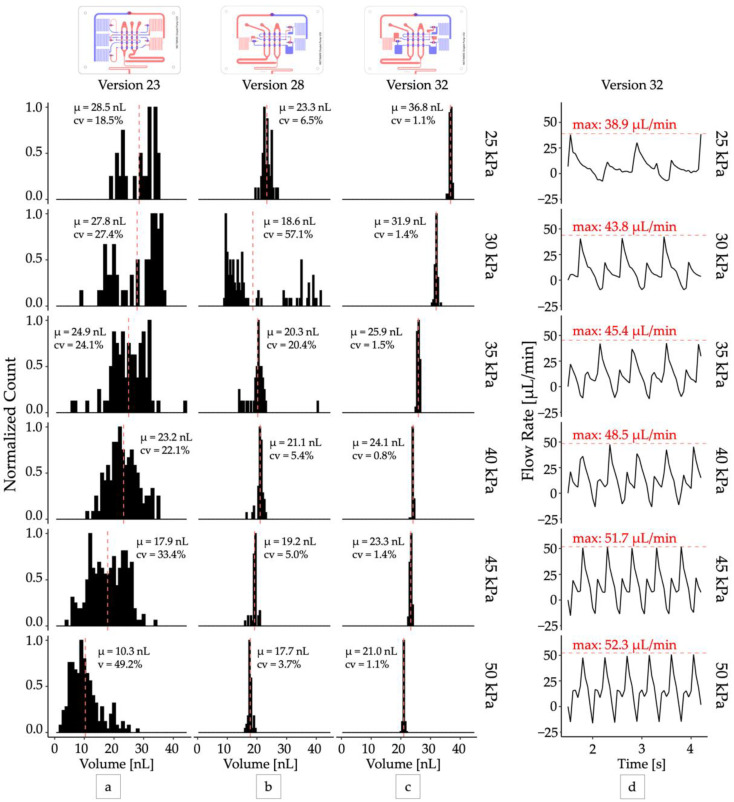
Optimizing droplet monodispersity. Distributions of droplet volumes produced by design iterations (**a**) Version 23: five-inverter two chamber pump, (**b**) Version 28: three-inverter single chamber pump with capacitors, (**c**) Version 32: three-inverter single chamber pump with larger capacitors. All distributions were sampled at steady state with *N* > 40. (**d**) The total output flow rate produced by the optimized droplet generator (version 32). The peak flow rate (red dashed lines) delivered by each pump cycle increased from 38.9 µL/min to 52.3 µL/min as operating vacuum increased from 25 kPa to 50 kPa.

**Figure 3 micromachines-13-01784-f003:**
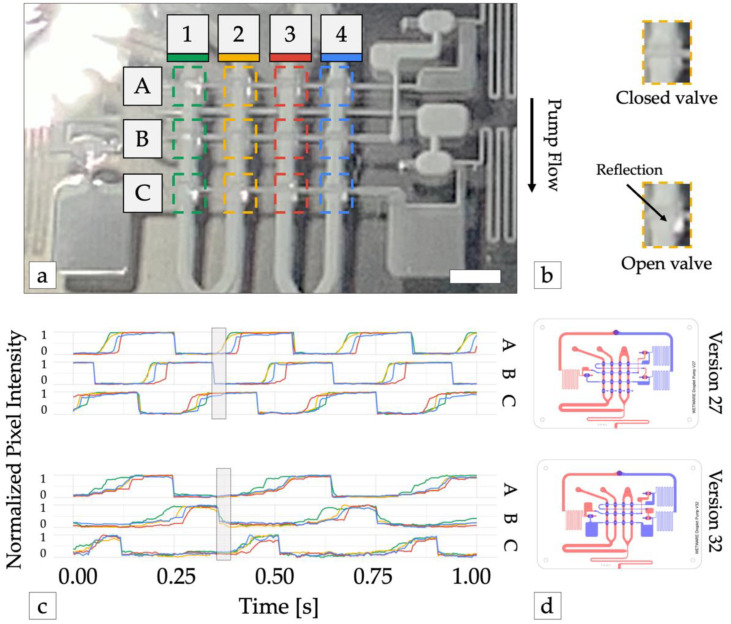
Optimizing pump waveform synthesis. (**a**) Pump operation was monitored with high-speed video. The reflection of incident light from the membrane of each pump valve was used to detect valve state (open or closed). Dashed colored lines show the area of the image analyzed for each valve. Scale bar = 3 mm. (**b**) Valve detail shows open valves reflect more light back the camera. (**c**) Time traces of normalized reflected light from each valve from device versions 27 (top) and version 32 (bottom). 1 = valve open (vacuum pressure), 0 = valve closed (atmospheric pressure). Adding large capacitors before and after each row of pump valves (version 32) increased the time required to open each row of valves and the subsequent inverter gate, preventing all pump stages (A, B, and C) from being open simultaneously (shaded rectangle), which resulted in improved pump performance. (**d**) Illustrations of device versions 27 and 32. Channels in the top and bottom layers are shown in blue and red, respectively.

**Figure 4 micromachines-13-01784-f004:**
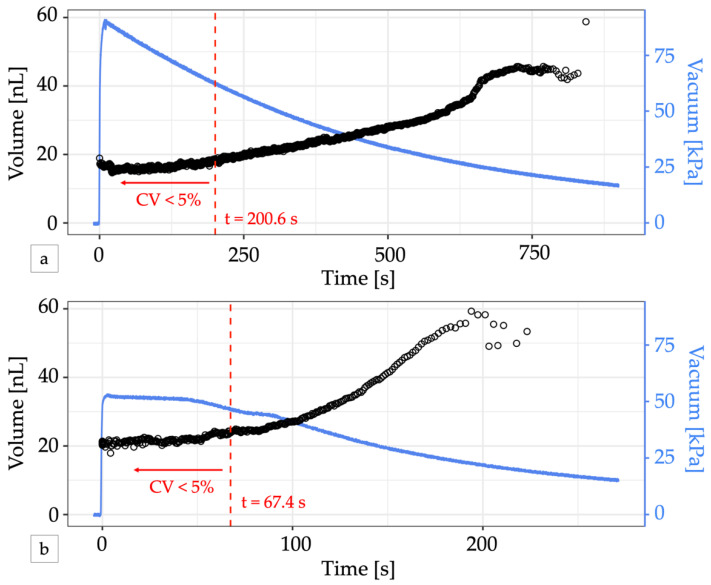
Electricity-free droplet generation. (**a**) A 60 mL locking syringe supplies enough vacuum pressure (blue line) to generate droplets (black circles) for over 10 min. The change in volume is minimal for vacuum pressures greater than 50 kPa. For the first 200.6 s (dashed red line), 465 droplets were produced with a CV < 5% and a mean volume of 16.52 nL. (**b**) Adding a vacuum regulator to the syringe output can provide a stable vacuum pressure for a short time but produced fewer uniform-sized droplets. In the first 67.4 s, 158 droplets with a mean volume of 21.77 nL were produced with a CV < 5% (dashed red line). In both graphs, the first two droplets generated were omitted from volume calculations to account for system stabilization.

## Data Availability

Not applicable.

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
