# Peer review of "Phase-Optimized Peristaltic Pumping by Integrated Microfluidic Logic"

_micromachines, 2022, doi:10.3390/mi13101784_

Round 1
Reviewer 1 Report
This paper presents a droplet generation system using integrated peristaltic pumps that are coordinated using microfluidic logic circuits. Though each particular technique is not new, the idea of combining these techniques for the generation of droplets using a small volume of samples with a syringe vacuum is very thoughtful. I like it. My comments would be relatively minor, but if the authors could revise the manuscript accordingly, this work could be fun and inspiring to the readers.
1. Could the authors present a frame-by-frame description of the working principle of the system? I think this will help the readers who are not familiar with microfluidic logic to understand.
2. I find it hard to follow the design iterations of such many versions. It would be nicer to summarize several design points, and perform a controlled experiment on each of the design points, to show the effectiveness of the design points in optimizing the performance of the device.
3. Could the authors add some information about the actual yield of the droplet generation, i.e., how much liquids are added to the inlet, and the material loss percentages?
Author Response
The authors appreciate the supportive and helpful comments from Reviewer 1. Responses to specific suggestions follow below.
(1) Could the authors present a frame-by-frame description of the working principle of the system? I think this will help the readers who are not familiar with microfluidic logic to understand.
Thank you for this suggestion. We have added an additional supplementary figure (Figure S2) depicting the ideal waveform generated by a 3-inverter oscillator pump. Additional detail on oscillator pumps, including still frames depicting the pump action, can be found in reference 18 (Duncan et al).
(2) I find it hard to follow the design iterations of such many versions. It would be nicer to summarize several design points, and perform a controlled experiment on each of the design points, to show the effectiveness of the design points in optimizing the performance of the device.
This is a good point. We have revised Supplementary Table 1 (Table S1) and added an additional column explaining some of the hypotheses tested and results observed.
(3) Could the authors add some information about the actual yield of the droplet generation, i.e., how much liquids are added to the inlet, and the material loss percentages?
We have added an approximate measure of the waste produced during the stabilization time. Also, a note was added to the discussion that the swept volume from external syringes tubing is eliminated.
Reviewer 2 Report
Erik et al. reported a droplet generation with a fast stabilization time by implementing on-chip pneumatic logic to control phase-optimized peristaltic pump. Could you please clarify below;
1. Do you have a video from before loading reagents till starting to generate droplet. That will be easier to understand the device working principle.
2. Some numbers in the graphs such as Fig.2 and Fig. 3 are too small.
3. In Fig.2, it is hard to understand what are the differences between version 23, 28, 32 without schematics in the manuscript.
4. In Supplemental 6, there is no figure for the vacuum source, which is misleading if only see the figure. As author explained in the figure caption and manuscript, I think it would be good to mention in the figure.
5. It is not clear which parameters affect the performance. For example in Fig.2, the author compared (1)version23: 5 inverters, 2 pump, (2)version 28: 3 inverters, 1 pump, (3)version 32, 3 inverters, 1 pump with capacitors. Why author did not change 1 parameter (such as change only the number of inverters) to see how it will affect the performance. By tuning multiple parameters at the same time, it is tough to understand the individual effect. Comparison between (2) and (3) make sense as only capacitor is changed.
6. As authors did lots of revisions and studies, I am wondering whether authors can provide further discussion such as how to select droplet configuration (cross-flow, co-flow etc[1]) and droplet generation methods (the author only mentioned active droplet generations are better than passive one. how about its details such as squeezing, dripping etc..) from their experiences.
[1] P. Zhu, L. Wang, Lab Chip, 2017, 17, 34-75.
Overall, the paper tackled an interesting topic. The authors showed their design revision records in the supplemental information, which I think is beneficial for readers to think of design. However, these revision records were not well discussed and reflected in the manuscript itself. Thus, some of figures are difficult to understand for me.
Author Response
The authors thank Reviewer 2 for providing detailed and insightful suggestions. Point-by-point responses are provided below.
(1) Do you have a video from before loading reagents till starting to generate droplet. That will be easier to understand the device working principle.
Unfortunately, we do not have any publication-quality videos showing the device priming. However, self-priming functionality was a goal of the design optimization, and this detail was added to the revised supplementary table (Table S1).

(2) Some numbers in the graphs such as Fig.2 and Fig. 3 are too small.
Thank you for pointing this out. We have enlarged the text in all figures.

(3) In Fig.2, it is hard to understand what are the differences between version 23, 28, 32 without schematics in the manuscript.
Thank you for this suggestion. We have added the schematics for devices 23, 28, and 32 to Figure 2.

(4) In Supplemental 6, there is no figure for the vacuum source, which is misleading if only see the figure. As author explained in the figure caption and manuscript, I think it would be good to mention in the figure.
This is a good point, and we agree. We have added labels for the vacuum source the figure (now Figure S7).

(5) It is not clear which parameters affect the performance. For example in Fig.2, the author compared (1)version23: 5 inverters, 2 pump, (2)version 28: 3 inverters, 1 pump, (3)version 32, 3 inverters, 1 pump with capacitors. Why author did not change 1 parameter (such as change only the number of inverters) to see how it will affect the performance. By tuning multiple parameters at the same time, it is tough to understand the individual effect. Comparison between (2) and (3) make sense as only capacitor is changed.
Thank you for pointing these out. You make a good point. We have revised Supplementary Table 1 (Table S1) and added an additional column explaining some of the hypotheses tested and results observed.

(6) As authors did lots of revisions and studies, I am wondering whether authors can provide further discussion such as how to select droplet configuration (cross-flow, co-flow etc[1]) and droplet generation methods (the author only mentioned active droplet generations are better than passive one. how about its details such as squeezing, dripping etc..) from their experiences.
[1] P. Zhu, L. Wang, Lab Chip, 2017, 17, 34-75.
This is indeed an interesting question, but droplet generation methods other than a T-junction were not explored in this work. The authors respectfully believe a comparison of droplet generation methods is beyond the scope of this paper.
Round 2
Reviewer 2 Report
Authors addressed all my concerns. I still believe it is better to pick one design and show step-by-step operation using schematics, so more people can understand its device working principle.